# Eu^3+^- and Tb^3+^-Based Coordination Complexes of Poly(*N*-Isopropyl,*N*-methylacrylamide-*stat*-*N,N*-dimethylacrylamide) Copolymer: Synthesis, Characterization and Property

**DOI:** 10.3390/polym14091815

**Published:** 2022-04-29

**Authors:** Jian Li, Guihua Cui, Siyuan Bi, Xu Cui, Yanhui Li, Qian Duan, Toyoji Kakuchi, Yougen Chen

**Affiliations:** 1Institute for Advanced Study, Shenzhen University, Shenzhen 518060, China; lijian0726@foxmail.com; 2College of Physics and Optoelectronic Engineering, Shenzhen University, Shenzhen 518060, China; 3Department of Chemistry, Jilin Medical University, Jilin 132013, China; cuiyuhan1981_0@sohu.com; 4Shenzhen Huizhi Technology Co., Ltd., Shenzhen 518102, China; bisiyuan0129@126.com; 5School of Materials Science and Engineering, Changchun University of Science and Technology, Changchun 130022, China; cuixu@cust.edu.cn (X.C.); liyanhui@ciac.ac.cn (Y.L.); duanqian88@hotmail.com (Q.D.); kakuchi@eng.hokudai.ac.jp (T.K.); 6Division of Applied Chemistry, Faculty of Engineering, Hokkaido University, Sapporo 060-8628, Japan

**Keywords:** group transfer polymerization, rare earth complexes, thermoresponsive, fluorescent property, cell viability

## Abstract

This contribution reports the syntheses, structural analyses and properties of europium (Eu^3+^)- and terbium (Tb^3+^)-based coordination complexes of poly(*N*-isopropyl,*N*-methylacrylamide-*stat*-*N,N*-dimethylacrylamide) (poly(iPMAm-*stat*-DMAm)) copolymer, named as poly-Eu(III) and poly-Tb(III), respectively. In greater detail, poly(iPMAm_85_-*stat*-DMAm_15_) is first prepared by random copolymerization of *N*-isopropyl,*N*-methylacrylamide (iPMAm) and *N,N*-dimethylacrylamide (DMAm) via group transfer polymerization (GTP). Next, poly(iPMAm_85_-*stat*-DMAm_15_) is used as the polymer matrix for chelating with Eu^3+^ and Tb^3+^ cations at its side amide groups, to produce poly-Eu(III) and poly-Tb(III). Their structural characterizations by FT-IR spectroscopy and XPS confirm the formation of polymeric complexes. The study on their fluorescence emission characteristics and luminescence lifetime demonstrates that Poly-Eu(III) shows four strong emission peaks at 578, 593, 622, and 651 nm, which are responsible for the electron transitions from the excited ^5^*D*_0_ state to the multiplet ^7^*F*_J_ (J = 0, 1, 2, 3) states, respectively, and poly-Tb(III) also displays four emission peaks at 489, 545, 588, and 654 nm, mainly due to the electron transitions of ^5^*D*_4_ → ^7^*F*_i_ (i = 6, 5, 4, 3). The luminescence lifetimes of poly-Eu(III) (τ_poly-Eu(III)_) and poly-Tb(III) (τ_poly-Tb(III)_) are determined to be 4.57 and 7.50 ms, respectively. In addition, in aqueous solutions, poly-Eu(III) and poly-Tb(III) are found to exhibit thermoresponsivity, with their cloud temperatures (*T*_c_s) locating around 36.4 and 36.8 °C, respectively. Finally, the cytotoxicity study on the human colon carcinoma cells LoVo and DLD1 suggests that the luminescent Eu^3+^ and Tb^3+^ in the chelated state with poly(iPMAm-*stat*-DMAm) show much better biocompatibility and lower toxicity than their inorganic salts.

## 1. Introduction

Luminescent rare earth complexes have become the research hotspot of new rare earth functional materials, because of their unique luminescence properties, such as extremely sharp emission bands [1,2], long excited-state lifetimes [3,4,5], and potential high internal quantum efficiency [6,7,8]. When rare earth metals are complexed with appropriate organic ligands, their improved solubility and dispersion would make their luminescence properties more prominent than they are in states of inorganic salts [9,10]. Furthermore, rare earth metals complexed with a polymeric matrix, namely, the so-called polymer–rare earth complexes, possess the combined properties of luminescent rare earth metals and the polymer matrix, making them have good solubility in solution, high mechanical strength in bulk, excellent processability, and outstanding luminescent emission properties. These features are of great significance for the potential applications in many fields, such as luminescent therapy probes in vivo, photo-driven catalysts, photoluminescent/electroluminescent polymer films, and active layers in solar cells [11,12,13]. Wang et al. [14] synthesized luminescent polymer-functionalized mesoporous SBA-16-type hybrid materials with encapsulated lanthanide (Eu^3+^ and Tb^3+^) complexes, which significantly improved the luminescent properties with the introduction of the organic ligand phen. Gao et al. [15] first synthesized a bidentate Schiff base ligand with side chains of polysulfone (PSF), then prepared polymer−rare earth complexes, PSF-(SB)_3_-Eu(III), and a ternary complex, PSF-(SB)_3_-Eu(III)-(Phen)_1_, through the coordination reaction, and the luminescent properties were greatly improved. Krsmanovic et al. [16] explored a synthesis route based on the polymer complex solution method for the production of rare earth-doped Lu_2_O_3_ crystalline nano powders. Xie et al. [17] reported a series of polymer–rare earth complexes with Eu^3+^, including binary complexes containing the single ligand poly(ethylene-co-acrylic acid) (EAA) and the ternary complexes, using 1,10-phenanthroline (phen), dibenzoylmethane (DBM) or thenoyltrifluoroacetone (TTA) as the second ligand, which possessed potential applications as luminescent materials.

For more than a decade, the application scope of polyacrylamide (PAm) and its derivatives, such as *N*-substituted and *N,N*-disubstituted PAms (*N*-PAms and *N,N*-PAms, respectively), has rapidly expanded to the fields of biomedical applications, including plastic and reconstructive applications [18,19], contact lenses [20,21], nucleic acid and protein separation [22,23], and drug delivery [24,25]. In addition, the use of PAm as a polymer matrix to synthesize luminescent polymer–rare earth complexes can fully combine the water solubility and thermoresponsive properties of PAm with rare earth ions, to improve their fluorescence properties [26,27,28,29]. However, *N*-PAms and *N,N*-PAms, with controlled molar masses, narrow dispersity, and well-defined structures, have rarely been used as polymer matrices to synthesize luminescent polymer–rare earth complexes. In this field, Duan et al. first synthesized D-glucosamine end-functionalized poly(*N*-isopropylacrylamide) (GA-PNIPAM) by atom transfer radical polymerization (ATRP), and studied the coordinating interaction with Eu(III), and the thermoresponsive and fluorescence properties of the Eu(III)/PNIPAM complexes [30,31].

Therefore, it is meaningful to expand the scope of PAms as polymer ligands for rare earth elements, to realize the functions derived from both components. In our previous work, we developed a new type of *N,N*-PAms, poly(*N*-isopropyl,*N*-methylacrylamide) (PiPMAm), by organocatalytic group transfer polymerization (GTP), and studied the thermoresponsive properties of its homopolymers and statistical and block copolymers, consisting of *N*-isopropyl,*N*-methylacrylamide (iPMAm) and *N,N*-dimethylacrylamide (DMAm) structural units [32]. The metal-free poly(iPMAm-*stat*-DMAm), with a controlled molar mass and narrow dispersity, showed quasi-controllable cloud temperature (*T*_c_). In this work, we chose europium (Eu^3+^) and terbium (Tb^3+^), which showed red and green emissions, respectively, as rare earth ions, and poly(iPMAm_85_-*stat*-DMAm_15_) as the polymer matrix to synthesize thermoresponsive luminescent polymer–rare earth complexes, named poly-Eu(III) and poly-Tb(III). The present contribution describes, in detail, (i) the syntheses and characterizations of poly-Eu(III) and poly-Tb(III) complexes, (ii) their fluorescent and thermoresponsive properties, and (iii) an assessment of their cell viability.

## 2. Experimental Section

### 2.1. Materials and Measurements

*N,N*-dimethylacrylamide (DMAm) was purchased from Tokyo Kasei Kogyo Co., Ltd., Tokyo, Japan (TCI) and used after distillation over CaH_2_ under reduced pressure. Tris(pentafluorophenyl)borane (B(C_6_F_5_)_3_) (TCI) was used after the recrystallization from *n*-hexane at −30 °C. Acryloyl chloride (TCI) was used after distillation at 77 °C. Acetone (>98%), deuterated chloroform (CDCl_3_, >99.8%), *n*-hexane (98%), tetrahydrofuran (THF, >99%), methanol (MeOH, >98%), toluene (99%), triethylamine (>99.0%), 3,5-dibromoaniline, triisopropylsilylacetylene, trimethylsilylacetylene and sodium hydride (60 wt.% in mineral oil) were purchased from TCI Chemicals. Dichloromethane (CH_2_Cl_2_, >99.5%; extra dry, with molecular sieves, water < 50 ppm) and tetrahydrofuran (THF, >99.5%; extra dry, with molecular sieves, water < 50 ppm) were purchased from Energy Chemical Co., Ltd., Shanghai, China. Toluene and THF were distilled over Na/benzophenone under an argon atmosphere and degassed by three freeze–pump–thaw cycles prior to use. All other chemicals were purchased from available suppliers and used without further purification.

Polymerization was carried out in an MIKROUNA stainless-steel glove box equipped with a gas purification system under a dry argon atmosphere (H_2_O, O_2_ < 0.01 ppm). The moisture and oxygen contents in the glove box were monitored by an MB-MO-SE 1 and an MB-OX-SE 1 sensor, respectively. The ^1^H and ^13^C NMR spectroscopy was recorded using a Bruker AVANCE III HD 400, Billerica, MA USA. Size exclusion chromatography (SEC) in DMF was performed at 40 °C at a flow rate of 0.35 mL min^−1^ using a TosohHLC-8320 GPC System equipped with two TSK gel Super Multipore HZ-M columns (4.6 mm I.D. × 15 cm × 2), an EcoSEC GPC System with an RI detector (+, 0.5 s), a UV-8320 detector (254 nm, +0.5 s), a GPC workstation EcoSEC-WS, and an auto sampler, to which 10 µL of sample was injected in a concentration of 0.2 wt.%. The number-average molecular weight (*M*_n,SEC_) and polydispersity index (*Ɖ*) of the polymers were calculated based on PMMA standards. FT-IR spectra were recorded on a Perkin Elmer Frontier, and samples fashioned into KBr pellet disks were placed into a holder for transmission IR spectral analysis. The fluorescence spectra were recorded using a fluorescence spectrophotometer (RF-5301PC, Shimadzu, Kyoto, Japan). Quantum yields were determined by comparison of the total light emitted from the solutions to the total light emitted from a known standard [Ru(bipy)_3_]Cl_2_ [33]. XPS spectra (Al Kα) were recorded with a Thermo Fisher Scientific K-Alpha instrument, Waltham, MA USA. The cloud point measurements were performed on the ultraviolet–visible (UV–vis) spectra by passing through a 10 mm path-length cell using a Jasco V-770 spectrophotometer equipped with a temperature controller (Jasco CTU-100, Tokyo, Japan). The hydrodynamic radii (*R*_h_s) of the obtained polymer in deionized water were analyzed using a dynamic light scattering (DLS) detector (Wyatt Technology, Dyna Pro Nanostar^®^, Santa Barbara, CA, USA). Cell viability was evaluated by MTT, and the optical density (OD) was measured at 490 nm with a microplate reader (Bio-Rad, Hercules, CA, USA). Cell viability was determined as a percentage of the negative control (untreated cells).

### 2.2. Synthesis of Statistical Copolymer Poly(iPMAm_85_-stat-PDMAm_15_)

iPMAm (246 μL, 1.7 mmol), DMAm (33 μL, 0.3 mmol), SKA^Et^ (40 μL, 20.0 μmol; 0.50 mol L^−1^ CH_2_Cl_2_), and CH_2_Cl_2_ (1.60 mL) were added to a test tube, followed by the addition of the B(C_6_F_5_)_3_ stock solution in CH_2_Cl_2_ (80 μL, 4.0 μmol; 0.05 mol L^−1^) for 2 h in a glove box. The polymerization was quenched by adding a small amount of a 2-PrOH/pyridine mixture to the polymerization solution. Aliquots were removed from the reaction mixture to determine the conversion of iPMAm and DMAm using ^1^H NMR measurements. The polymer product was purified by dialysis against methanol using a cellophane tube, after which the product was lyophilized to give poly(iPMAm_85_-*stat*-PDMAm_15_) as a white solid. Yield, 244.8 mg (98.2%); number-average molecular weight (*M*_n,SEC_), 13.0 kg mol^−1^; polydispersity index (*Ɖ*), 1.17.

### 2.3. Synthesis of Poly-Eu(III) and Poly-Tb(III) Complexes

A typical method for the synthesis of poly-Eu(III) and poly-Tb(III) complexes is described as follows: EuCl_3_ (100.0 mg, 3.87 mmol), poly(iPMAm_85_-*stat*-PDMAm_15_) (1.0 g, 0.07 mmol), and ethanol (15 mL) were added into a flask, then stirred at 40 °C for 24 h. The product was purified by dialysis against methanol using a cellophane tube (Spectra/Por 6 Membrane; MWCO: 1000), after which the product was lyophilized to produce the poly-Eu(III) complex as a white solid.

### 2.4. Determination of the Cloud Point (T_c_)

An aqueous solution of the polymer (1 wt.%) was sonicated for several minutes, and the resulting clear solution was then transferred to a 10 mm quartz cell. The transmittance of the aqueous solution at 500 nm was recorded by a UV–vis spectrophotometer equipped with a temperature controller. The solution was heated at a heating rate of 1.0 °C min^−1^.

### 2.5. Determination of Hydrodynamic Radius (R_h_)

A sonicated aqueous solution of the polymer (1 wt.%) was filtrated into glass cells using a 0.45 μm PTFE filter. The relaxation time (*τ*) distribution and particle size distribution were obtained by the CONTIN analysis of the autocorrelation function. The apparent diffusion coefficients *D* were calculated using the following equation:$${\frac{\Gamma }{q^{2}}|}_{q\;\to \;0}=D$$
where Γ is the relaxation frequency (Γ = *τ*^−1^) and *q* is the wavevector defined by the following equation:$$q=\frac{4\pi n}{\lambda }\;\sin\left(\frac{\theta }{2}\right)$$
where *λ* is the wavelength of the laser beam (532 nm), *θ* is the scattering angle, and *n* is the refractive index of the media. Consequently, the hydrodynamic radius (*R*_h_) was calculated from the Stokes–Einstein relation as follows:$$R_{h}=\frac{k_{B}T}{6\pi \eta \Gamma }q^{2}=\frac{k_{B}T}{6\pi \eta D}$$
where *k*_B_ is the Boltzmann constant, *T* is the temperature, and *ƞ* is the viscosity of the medium.

### 2.6. Cell Viability Study

Typically, cell viability was investigated using the human colon carcinoma cells LoVo and DLD1 in culture. After incubation for 24 h in 96-well plates (8 × 10^4^ cells mL^−1^ per well) using Dulbecco’s modified Eagles medium (DMEM) in an incubator (36 °C, 5% CO_2_), the culture medium was mixed with 200 μL of DMEM containing a sample of poly(iPMAm_85_-*stat*-PDMAm_15_), poly-Eu(III), poly-Tb(III), EuCl_3_ and TbCl_3_ under concentrations from 0.1 to 1000.0 μg mL^−1^. The mixture was further incubated for 48 h [34]. Each sample was tested in five replicates per plate, then 20 µL of MTT solution was added to the mixture in each well, which was incubated for an additional 4 h. Next, 200 μL of DMSO was added and the mixtures were shaken at room temperature. Five replicate wells were used for the control and test concentrations for each microplate. The cell viability (%) was calculated by the following equation:Cell viability (%) = (A_sample_/A_control_) × 100%
where A_sample_ was the absorbance of the cells incubated in DMEM and mixture, and A_control_ was the absorbance of the cells incubated in DMEM [35].

## 3. Results and Discussion

### 3.1. Synthesis of Poly-Eu(III) and Poly-Tb(III)

For the preparation of poly-Eu(III) and poly-Tb(III), we first synthesized poly(iPMAm_85_-*stat*-DMAm_15_) as the polymer matrix, according to the previously reported B(C_6_F_5_)_3_-catalyzed GTP method, using 1-methoxy-2-methyl-1-(triethylsilyloxy)propene (SKA^Et^) as the initiator in CH_2_Cl_2_, under the condition of [iPMAm]_0_/[DMAm]_0_/[SKA^Et^]_0_/[B(C_6_F_5_)_3_] = 85/15/1/0.2, as shown in Figure 1.

The monomers, iPMAm and DMAm, were completely consumed after 2 h, and the structure of the obtained polymer was verified using ^1^H NMR, as shown in Figure 1. The methyl and methoxy protons, as the α-terminal group of the polymer, were observed at 1.16–1.20 (signal a) and 3.67 ppm (signal b), respectively. The signals due to the methyl protons of the *N*-isopropyl group in PiPMAm (signal f) and the *N*-methyl group in PDMAm (signal h) were observed at 0.98–1.20 and 2.81–3.19 ppm, respectively. The methenyl protons of the *N*-isopropyl group in PiPMAm (signal g) were observed at 4.69–4.92 ppm. The monomer composition in the copolymer could be regarded as the same as the monomer feed ratio of 85/15, due to the quantitative monomer conversion confirmed by the ^1^H NMR spectrum, i.e., the obtained copolymer was poly(iPMAm_85_-*stat*-DMAm_15_).

The SEC trace of poly(iPMAm_85_-*stat*-DMAm_15_) exhibited a unimodal distribution, as shown in Figure 2. The SEC-determined number-average molar mass (*M*_n,SEC_) and dispersity (*Ɖ*) were 13.0 kg mol^−1^ and 1.17, respectively. All the results confirm the desired synthesis of this polymer matrix.

Poly-Eu(III) and poly-Tb(III) were prepared by coordination reactions between poly(iPMAm_85_-*stat*-DMAm_15_) and the rare earth chlorides EuCl_3_ and TbCl_3_, respectively. The structural analyses of poly-Eu(III) and poly-Tb(III) were first commenced by FT-IR spectroscopy. Figure 3 shows the FT-IR spectra of poly(iPMAm_85_-*stat*-PDMAm_15_), poly-Eu(III), and poly-Tb(III). For poly(iPMAm_85_-*stat*-PDMAm_15_), the characteristic adsorptions due to the stretching and bending vibrations of the acylamino group (*ν*_N-H_ and δ_N-H_) and stretching vibration of carbonyl (ν_C=O_) appear at 3301, 1550, and 1674 cm^−1^, respectively. In comparison, the same bond vibrations in poly-Eu(III) are observed at 3293, 1538, and 1657 cm^−1^, and those for poly-Tb(III) appear at 3295, 1540, and 1658 cm^−1^. The redshift of these characteristic absorptions after the coordination reactions resulted from the formation of coordination bonds between Eu^3+^/Tb^3+^ and the carbonyl groups in poly(iPMAm_85_-*stat*-PDMAm_15_). Namely, part of the lone pair electrons of the O atom transfers to the outer orbitals of the rare earth ion, which weakens the σ covalent bond and force constant of the C=O. In addition, the inductive effect along the NH-C=O → Eu^3+^/Tb^3+^ also lowers the electron density of the N atom, leading to the redshift phenomenon [36].

The structural characterizations of poly-Eu(III) and poly-Tb(III) were further implemented by X-ray photoelectron spectroscopy (XPS). Figure 4a shows the XPS profiles of poly(iPMAm_85_-*stat*-PDMAm_15_), poly-Eu(III), and poly-Tb(III), and their average binding energies of O 1s, N 1s, Eu 4d, and Tb 4d are listed in Table 1. Poly(iPMAm_85_-*stat*-DMAm_15_) shows peaks at 283.71, 530.51, and 399.12 eV, corresponding to the C 1s, N 1s, and O 1s, respectively, while, in addition to these peaks, poly-Eu(III) and poly-Tb(III) display additional Eu 4d and Tb 4d peaks at 139.68 and 145.12 eV, respectively. The average binding energies of O 1s and N 1s in poly-Eu(III) are 531.25 and 399.58 eV, respectively, showing a +0.74 and +0.46 eV shift, in comparison with those in poly(iPMAm_85_-*stat*-PDMAm_15_). Meanwhile, the average binding energy of Eu 4d in poly-Eu(III) (139.68 eV) is 5.44 eV smaller than that of EuCl_3_ (145.12 eV). The change in average binding energy before and after the coordination reaction between poly(iPMAm_85_-*stat*-PDMAm_15_) and TbCl_3_ shows the similar tendency, i.e., the average binding energies of O 1s and N 1s in poly-Tb(III) are +0.70 and +0.32 eV greater than those in poly(iPMAm_85_-*stat*-PDMAm_15_), respectively, while the average binding energy of Tb 4d in poly-Tb (148.97 eV) is 2.86 eV smaller than that of TbCl_3_ (151.83 eV). In summary, the increase in the average binding energies of O 1s and N 1s of poly-Eu(III) and poly-Tb(III) is caused by the decrease in electron density for the O 1s and N 1s atoms. On the contrary, the decrease in the average binding energy of Eu 4d and Tb 4d in the polymer–rare earth complexes is due to the enhanced electron density around the rare earth metal cations after the lone pair electrons of O and N atoms of the acylamino group are coordinated to the outer orbital of Eu^3+^ or Tb^3+^. These results strongly indicate the formation of the target poly-Eu(III) and poly-Tb(III) complexes [37].

### 3.2. Luminescence Properties of Poly-Eu(III) and Poly-Tb(III)

The luminescence properties of the poly-Eu(III) and poly-Tb(III) complexes are shown in Figure 5. The excitation spectrum of poly-Eu(III) exhibits wide-range absorption peaks from 350 to 472 nm, and the peak at 355 nm shows the maximum excitation intensity, while only a negligible excitation peak is observed for EuCl_3_ in all the excitation fluorescence spectra (Figure 5a). This phenomenon can be attributed to the π-π* transition by exciting the carbonyl and amide groups of the complexes. In addition, the conjugated structure increases the electron delocalization and the absorption of ultraviolet light [33], which makes the intensity of the excitation peak increase sharply. Figure 5b indicates that EuCl_3_ exhibits very weak emission peaks, but poly-Eu(III) gives four strong emission peaks at 578, 593, 622, and 651 nm, which are responsible for the transition from an excited ^5^*D*_0_ state to multiplet ^7^*F*_J_, (J = 0, 1, 2, 3) states, respectively. Moreover, the 4*f* orbital is shielded by the outer shell of the 5*s* and 5*p* orbitals, and the *f*-*f* absorption bands are very narrow, which makes the intensity of the emission peak at 622 nm (^5^*D*_0_ → ^7^*F*_2_) 13.2 times that of EuCl_3_.

The excitation spectra of the poly-Tb(III) complex and TbCl_3_ are shown in Figure 5c. Poly-Tb(III) displays wide-range absorption from 280 to 390 nm, and the maximum peak appears at 350 nm. When compared with poly-Tb(III), TbCl_3_ shows very weak absorption, mainly due to the extremely low solubility of TbCl_3_ in water. Figure 5d is the emission spectra of poly-Tb(III) and TbCl_3_. Similarly to poly-Eu(III), poly-Tb(III) has four emission peaks at 489, 545, 588, and 654 nm, due to the transitions of ^5^*D*_4_ → ^7^*F*_i_ (i = 6, 5, 4, 3). The intensity at the maximum emission (545 nm) of poly-Tb(III), due to the ^5^*D*_4_ → ^7^*F*_5_ transition, is 13.4 times that of TbCl_3_, which originated from the transitions between the 4*f* states in poly-Tb(III).

The fluorescent lifetimes of the poly-Eu(III) and poly-Tb(III) complexes are calculated according to the luminescence decay curves in Figure 6. Poly-Eu(III) and poly-Tb(III) have τ_poly-Eu(III)_ = 4.57 ms and τ_poly-Tb(III)_ = 7.50 ms, respectively. The photoemission efficiency of the two rare earth complexes is increased, because the much higher coordination ability of the poly(iPMAm_85_-*stat*-DMAm_15_) matrix than the chloride anion further stabilizes the Eu^3+^ or Tb^3+^. The more stable coordination in the rare earth complexes can largely enhance the absorption coefficient. Therefore, the initial strong absorption of ultraviolet energy excites the amide ligand to the excited singlet (S1) state, and the transition from the S1 state to the triplet (T) state, and then the energy transfers non-radiatively from the lowest triplet state of the ligand to the resonance state of Eu^3+^ or Tb^3+^ [38,39]. The energy undergoes multiphoton relaxation and subsequent emission in the visible light region.

### 3.3. Thermoresponsive Properties

The thermal phase transition behavior of poly(iPMAm_85_-*stat*-DMAm_15_), poly-Eu(III), and poly-Tb(III) is studied by a UV–vis spectrophotometer and dynamic light scattering (DLS) measurements. The temperature at light transmittance of 1 wt.% polymer aqueous solution = 50% on the transmittance–temperature curve is used as the cloud temperature (*T*_c_). DLS is used to monitor the polymer assembly state, by determining the hydrodynamic radius (*R*_h_) of a polymer in the same aqueous solution. Figure 7 shows the transmittance dependence of poly(iPMAm_85_-*stat*-DMAm_15_), poly-Eu(III), and poly-Tb(III) on the temperature. Each transmittance–temperature curve shows a sharp transmittance decrease at 32–40 °C, indicating that the polymer undergoes a quick thermal phase transition. The *T*_c_s of poly(iPMAm_85_-*stat*-DMAm_15_), poly-Eu(III), and poly-Tb(III) are 36.1, 36.4, and 36.8 °C, respectively (Table 2). It seems that the complexes have a slightly higher *T*_c_ than poly(iPMAm_85_-*stat*-DMAm_15_), but the difference in *T*_c_ is so small that we cannot exclude the possibility that it may be caused by experimental error. In general, the incorporation of highly hydrophilic metal cations into thermoresponsive polymers would increase the *T*_c_ value.

The *R*_h_ values of poly(iPMAm_85_-*stat*-DMAm_15_), poly-Eu(III), and poly-Tb(III) in Table 2 are 3.1, 5.2, and 6.9 nm, respectively, at 28 °C lower than *T*_c_. In this case, all of them should be in a unimolecular hydrated state. It is interesting that poly-Eu(III) and poly-Tb(III) have much greater *R*_h_ values than the parent poly(iPMAm_85_-*stat*-DMAm_15_). It is rational to assume that poly-Eu(III) and poly-Tb(III) incorporated with rare earth metal cations, in a sense, are polyelectrolytes, which have stretched chain structures and, thus, hydrated dimensions, i.e., greater *R*_h_ values. When the temperature is enhanced to 42 °C higher than *T*_c_, the *R*_h_ values are 1758.4, 4216.3, and 3952.5 nm, respectively. Undoubtedly, the polymer chain undergoes a coil-to-globule transition from the hydrated unimolecular state at 28 °C to the dehydrated aggregated state at 42 °C. It should be noted that the *R*_h_ size of poly(iPMAm_85_-*stat*-DMAm_15_) at 42 °C was almost the same in our previous work [32], while poly-Eu(III) and poly-Tb(III) had a 2.4 times larger *R*_h_ than poly(iPMAm_85_-*stat*-DMAm_15_). This phenomenon could also be caused by the polyelectrolyte nature of poly-Eu(III) and poly-Tb(III) [40,41].

### 3.4. Assessment of Cell Viability

Since the *T*_c_ values of poly-Eu(III) and poly-Tb(III) are very close to the human body temperature, they are of great potential to be used as drug delivery systems (DDSs). Precisely for this, the assessment of cell viability is further carried out to evaluate their cytotoxicity for DDS materials. LoVo (CCL-229^TM^) is a cell line isolated in 1971 from the large intestine of a 56-year-old male, Caucasian, grade IV Dukes C colorectal cancer patient. LoVo cells can be used for cancer, toxicology, and immuno-oncology research, and high-throughput screening. DLD1 (CCL-221^TM^) is a colorectal adenocarcinoma cell line isolated from the large intestine of a colon adenocarcinoma patient. It can be used for cancer research. The cytotoxicity towards the human colon carcinoma cells LoVo and DLD1 is studied by comparing the biocompatibility of poly(iPMAm_85_-*stat*-DMAm_15_), poly-Eu(III), and poly-Tb(III), as well as EuCl_3_ and TbCl_3_. The percentage of cell viability is determined by comparing them with the control group, in which the cells are not exposed to polymer samples. When exposed to EuCl_3_ and TbCl_3_, the cell viability of LoVo and DLD1 drops sharply, as shown in Figure 8. Furthermore, the lethality resulting from EuCl_3_ and TbCl_3_ increases when increasing the salt concentration from 0.1 to 1000.0 μg mL^−1^, indicating a dose-dependent cytotoxic effect. In contrast to the rare earth chlorides, poly(iPMAm_85_-*stat*-DMAm_15_), poly-Eu(III), and poly-Tb(III) turn out to be non-toxic in both LoVo and DLD1 cells, with nearly 100% vitality in a wide concentration range (0.1 to 1000.0 μg mL^−^^1^), and no obvious difference in cytotoxicity is observed between the parent polymer matrix and its complexes. These results suggest that the complexation of a polymer matrix, poly(iPMAm_85_-*stat*-DMAm_15_), and Eu^3+^ or Tb^3+^ greatly reduces the toxicity of inorganic rare earth chlorides. These results confirm that the coordination of poly(iPMAm_85_-*stat*-DMAm_15_) with Eu^3+^ and Tb^3+^ to form complexes can enhance the biocompatibility and eliminate the cell. These new types of complexes are potential luminescent probing materials for targeted applications in cytology and immune omics, and anti-tumor therapy.

## 4. Conclusions

Poly-Eu(III) and poly-Tb(III) complexes are synthesized by a coordination reaction between poly(iPMAm_85_-*stat*-DMAm_15_) and their parent chloride salts. The complexes have coordination interactions between the O and N atoms of the acylamino group with Eu^3+^ and Tb^3+^, which bring about the strong emission peaks at 622 and 545 nm, respectively. Poly-Eu(III) and poly-Tb(III) have *T*_c_s near the human body temperature and slightly higher than that of the parent polymer matrix. The assessment of cell viability verifies that poly-Eu(III) and poly-Tb(III) can greatly enhance the biocompatibility and reduce the cell toxicity. These PAm rare earth complexes are expected to be used as luminescent probes in the biomedical field.

## Data Availability

Not applicable.

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
