# Peer review of "Eu^3+^- and Tb^3+^-Based Coordination Complexes of Poly(*N*-Isopropyl,*N*-methylacrylamide-*stat*-*N,N*-dimethylacrylamide) Copolymer: Synthesis, Characterization and Property"

_polymers, 2022, doi:10.3390/polym14091815_

Round 1

Reviewer 1 Report

This paper reported synthesis, characterization and property of new copolymer based on N-isopropyl,N-methylacrylamide and N,N-dimethylacryla-mide. Eu(III) and Tb(III) were complexing by this copolymer, and then photophysical properties have been studied. Complexes of lanthanide are relevant from the point of view of sensing ability. It is an interesting article, I definitely recommend it for publication, but after revision. There are some suggestions and comments:

1) There are two important parameter for polymer in aqueous solution: critical concentration of micellization and cloud point. Unfortunately, there is no information about first. Aggregation behavior of polymer depends on ionic strength of solution (actually, d- and f-metal cation). It wasn’t discussed.

2) It is not clear, why authors used mixed solution (EtOH+H2O). Coordination polymers of lanthanides with luminescence properties are very interesting in aqueous solution. If poly-Tb(III) and poly-Eu(III) are fluorescent in water, it should be shown, because it is difficult to sensitize fluorescent of Eu in aqueous solution.

3) It is not clear for me «low solubility and strong aggregation feature is a reason for weak absorption of TbCl3» (page 8).

4) It is confusing me that different experiments were done in different solution: luminescence decay in chloroform, Excitation and emission spectra in mixed solution (EtOH+H2O), temperature sensitive response in water.

5) Is there an opportunity to centrifuge an redisperse poly-Tb(III) and poly-Eu(III) from mixed solution?

Reviewer 2 Report

The manuscript entitled “Eu3+ and Tb3+ Based Coordination Complexes of Poly(N-isopropyl, N-methylacrylamide-stat-N,N-dimethylacrylamide) copolymer: Synthesis, Characterization and Property” by  Yougen Chen et al. describes the preparation and characterization of coordination polymers based on poly(N-isopropyl, N-methylacrylamide )-containing copolymer and lanthanide ions (Eu3+ and Tb3+).  The authors investigated the structure of the complexes by FTIR, XPS and fluorescence spectroscopy. The thermoresponsive behavior of the metal-based copolymer possessing N-isopropyl-N-methylacrylamide units was highlighted by DLS technique. Moreover, the cytotoxicity against some cancer cell lines was also evaluated.

The manuscript is appropriately presented. Some information still is needed to complete it:

  • The motivation for complexation with lanthanide ions is still unclear. Did authors try to obtain complexes with transition metal ions for biological interests: Cu, Fe, Ni, Co? In this idea, the authors should develop the state-of-the-art of PNIPAAm-based ligands and their metal complexes.
  • The authors must introduce a scheme of metal complexes, because the lanthanide ions exhibit a large coordination number and it is difficult to understand what and how many coordination groups are involved in the coordination of Eu(III) and Tb(III) ions.
  • The FTIR characterization is based on some assumptions that need to be confirmed. Thus, a schematic representation of the coordinative bonds into the complex would help more the IR assignments.
  • Also, a study of the complexation reaction in solution must be performed in order to establish the stoichiometry of the complex. This experiment also would help the estimation of the complex structure. Even based on XPS studies the authors should indicate a structure for the complex.
  • The electronic absorption spectra are also need to be presented, not only the fluorescence ones.
  • In Figure 5, there are no differences in the emission of the metal salt and metal complexes, only on the intensity. What was the motivation for emission tests?

The cell viability studies were performed only on tumor cell lines. The authors should mention the assignments of the cells, not only the abbreviation. Why the viability has not be tested on the normal cell lines? How important would this activity be if it were similar on normal cells? Since I think they need additional experiments/ explanations, that is why it would need minor revisions.

Round 2

Reviewer 1 Report

The authors have addressed all the concerns. I recommend its publication at its current version.